# BLACK BOX RECURSIVE TRANSLATIONS FOR MOLECULAR OPTIMIZATION

## ABSTRACT

Machine learning algorithms for generating molecular structures offer a promising new approach to drug discovery. We cast molecular optimization as a translation problem, where the goal is to map an input compound to a target compound with improved biochemical properties. Remarkably, we observe that when generated molecules are iteratively fed back into the translator, molecular compound attributes improve with each step. We show that this finding is invariant to the choice of translation model, making this a "black box" algorithm. We call this method Black Box Recursive Translation (BBRT), a new inference method for molecular property optimization. This simple, powerful technique operates strictly on the inputs and outputs of any translation model. We obtain new state-of-the-art results for molecular property optimization tasks using our simple drop-in replacement with well-known sequence and graph-based models. Our method provides a significant boost in performance relative to its non-recursive peers with just a simple "for" loop. Further, BBRT is highly interpretable, allowing users to map the evolution of newly discovered compounds from known starting points.

## 1 INTRODUCTION

Automated molecular design using generative models offers the promise of rapidly discovering new compounds with desirable properties. Chemical space is large, discrete, and unstructured, which together, present important challenges to the success of any molecular optimization campaign. Approximately $10^8$ compounds have been synthesized (Kim et al., 2015) while the range of potential drug-like candidates is estimated to between $10^{23}$ and $10^{80}$ (Polishchuk et al., 2013). Consequently, new methods for intelligent search are paramount.

A recently introduced paradigm for compound generation treats molecular optimization as a translation task where the goal is to map an input compound to a target compound with favorable properties (Jin et al., 2019b). This framework has presented impressive results for constrained molecular property optimization where generated compounds are restricted to be structurally similar to the source molecule.

We extend this framework to unconstrained molecular optimization by treating inference, vis-à-vis decoding strategies, as a first-class citizen. We observe that generated molecules can be repeatedly fed back into the model to generate even better compounds. This finding is invariant to the choice of translation model, making this a "black box" algorithm. This invariance is particularly attractive considering the recent emphasis on new molecular representations (Gómez-Bombarelli et al., 2018; Jin et al., 2018; Dai et al., 2018; Li et al., 2018; Kusner et al., 2017; Krenn et al., 2019). Using our simple drop-in replacement, our method can leverage these recently introduced molecular representations in a translation setting for better optimization.

We introduce Black Box Recursive Translation (BBRT), a new inference method for molecular property optimization. Surprisingly, by applying BBRT to well-known sequence- and graph-based models in the literature, we can produce new state-of-the-art results on property optimization benchmark tasks. Through an exhaustive exploration of various decoding strategies, we demonstrate the empirical benefits of using BBRT. We introduce simple ranking methods to decide which outputs are fed back into the model and find ranking to be an appealing approach to secondary property optimization. Finally, we demonstrate how BBRT is an extensible tool for interpretable and user-centric molecular design applications.

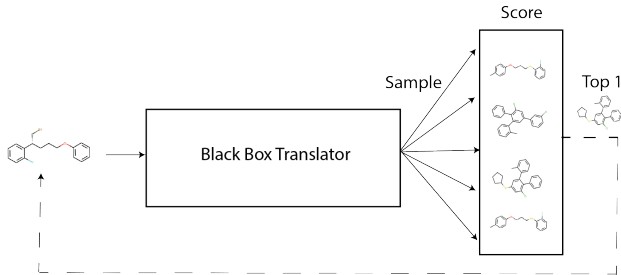

Figure 1: Black Box Recursive Translation (BBRT).

## 2 RELATED WORK

***De novo* molecular design**. Recent work has focused on learning new molecular representations including graphs (You et al., 2018b; Li et al., 2018), grammars (Kusner et al., 2017; Dai et al., 2018), trees (Jin et al., 2018), and sequences (Gómez-Bombarelli et al., 2018; Krenn et al., 2019). Provided with a molecular representation, latent variable models (Kusner et al., 2017; Dai et al., 2018; Jin et al., 2018)), Markov chains (Seff et al., 2019), or autoregressive models (You et al., 2018b) have been developed to learn distributions over molecular data. Molecular optimization has been approached with reinforcement learning (Popova et al., 2018; Zhou et al., 2019) and optimization in continuous, learned latent spaces (Gómez-Bombarelli et al., 2018). For sequences more generally, Mueller et al. (2017) perform constrained optimization in a latent space to improve the sentiment of source sentences.

We build on recent work introducing a third paradigm for design, focusing on molecular optimization as a translation problem (Jin et al., 2019b), where molecules are optimized by translating from a source graph to a target graph. While retaining high similarity, the target graph improves on the source graph's biochemical properties. With many ways to modify a molecule, Jin et al. (2019b) use stochastic hidden states coupled with a left-to-right greedy decoder to generate multi-modal outputs. We extend the translation framework from similarity-constrained molecular optimization to the unconstrained setting of finding the *best* scoring molecules according to a given biochemical property. We show that while their translation model is not fundamentally limited to constrained optimization, their inference method restricts the framework's application to more general problems.

**Matched molecular pair (MMP) analysis**. MMP analysis is a popular cheminformatics framework for analyzing structure-property relationships (Kramer et al., 2014; Turk et al., 2017). MMPs are extracted by mining databases of measured chemical properties and identifying pairs of chemical structures that share the same core and differ only by a small, well-defined structural difference; for example, where a methyl group is replaced by an isopropyl group (Tyrchan & Evertsson, 2017). The central hypothesis underlying MMPs is that the chemical properties of a series of closely related structures can be described by piecewise-independent contributions that various structural adducts make to the properties of the core.

MMP analysis has become a mainstream tool for interpreting the complex landscape of structure-activity relationships via simple, local perturbations that can be learnt and potentially transferred across drug design projects (Kubinyi, 1988). This framework serves as the chemistry analogue to an interpretability technique in machine learning called local interpretable model-agnostic explanations (LIME) (Ribeiro et al., 2016). Both MMP and LIME learn local surrogate models to explain individual predictions.

We view molecular translation as a learned MMP analysis. While Jin et al. (2019b) use neural networks to learn a single high-dimensional MMP step, we extend this framework to infer a sequence of MMPs, extending the reach of this framework to problems beyond constrained optimization.

**Translation models.** Machine translation models composed of end-to-end neural networks (Sutskever et al., 2014) have enjoyed significant success as a general-purpose modeling tool for many applications including dialogue generation (Vinyals & Le, 2015) and image captioning (Vinyals et al., 2015). We focus on a recently introduced application of translation modeling, one of molecular optimization (Jin et al., 2019b).

The standard approach to inference – approximately decoding the most likely sequence under the model – involves a left-to-right greedy search, which is known to be highly suboptimal, producing

generic sequences exhibiting low diversity (Holtzman et al., 2019). Recent work propose diverse beam search (Li & Jurafsky, 2016; Vijayakumar et al., 2018; Kulikov et al., 2018), sampling methods geared towards open-ended tasks (Fan et al., 2018; Holtzman et al., 2019), and reinforcement learning for post-hoc secondary objective optimization (Wiseman et al., 2018; Shen et al., 2016; Bahdanau et al., 2017). Motivated by molecular optimization as a translation task, we develop Black Box Recursive Translation (BBRT). We show BBRT generates samples with better molecular properties than its non-recursive peers for both deterministic and stochastic decoding strategies.

## 3 MOLECULAR OPTIMIZATION AS A TRANSLATION PROBLEM.

For illustrative purposes, we describe the translation framework and the corresponding inference method for sequences. We emphasize that our inference method is a black box, which means it is invariant to specific architectural and representational choices.

**Background.** Our goal is to optimize the chemical properties of molecules using a sequence-based molecular representation. We are given $(x, y) \in (\mathcal{X}, \mathcal{Y})$ as a sequence pair, where $x = (x_1, x_2, \ldots, x_m)$ is the source sequence with $m$ tokens and $y = (y_1, y_2, \ldots, y_n)$ is the target sequence with $n$ tokens, and $\mathcal{X}$ and $\mathcal{Y}$ are the source and target domains respectively. We focus on problems where the source and target domains are identical. By construction, our training pairs $(x, y)$ have high chemical similarity, which helps the model learn local edits to $x$. Additionally, $y$ always scores higher than $x$ on the property to be optimized. These properties are specified beforehand when constructing training pairs. A single task will typically optimize a single property such as potency, toxicity, or lipophilic efficiency.

A sequence to sequence (Seq2Seq) model (Sutskever et al., 2014) learns parameters $\theta$ that estimate a conditional probability model $P(y|x; \theta)$, where $\theta$ is estimated by maximizing the log-likelihood:

$$L(\theta) = \sum_{(x,y) \in (\mathcal{X}, \mathcal{Y})} \log P(y|x, \theta) \tag{1}$$

The conditional probability is typically factorized according to the chain rule: $P(y|x; \theta) = \prod_{t=1}^{n} P(y_t|y_{<t}, x, \theta)$. These models use an encoder-decoder architecture where the input and output are both parameterized by recurrent neural networks (RNNs). The encoder reads the source sequence $x$ and generates a sequence of hidden representations. The decoder estimates the conditional probability of each target token given the source representations and its preceding tokens. The attention mechanism (Bahdanau et al., 2014) helps with token generation by focusing on token-specific source representations.

## 4 BLACK BOX RECURSIVE TRANSLATION

### 4.1 CURRENT INFERENCE METHODS

For translation models, the inference task is to compute $y^* = \arg\max p(y|x, \theta)$. Because the search space of potential outputs is large, in practice we can only explore a limited number of sequences. Given a fixed computational budget, likely sequences are typically generated with heuristics. Decoding methods can be classified as deterministic or stochastic. We now describe both classes of decoding strategies in detail. For this section, we follow the notation described in Welleck et al. (2019).

**Deterministic Decoding.** Two popular deterministic methods include greedy search and beam search (Graves, 2012; Boulanger-Lewandowski et al., 2013). Greedy search performs a single left-to-right pass, selecting the most likely token at each time step: $y_t = \arg\max p(y_t|y_{<t}, x, \theta)$. While this method is efficient, it leads to suboptimal generation as it does not take into account the future when selecting tokens.

Beam search is a generalization of greedy search and maintains a set of $k$ hypotheses at each time-step where each hypothesis is a partially decoded sequence. While in machine translation beam search variants are the preferred method, for more open-ended tasks, beam search fails to generate diverse candidates. Recent work has explored diverse beam search (Li & Jurafsky, 2016; Vijayakumar et al., 2018; Kulikov et al., 2018). These methods address the reduction of number of duplicate sequences to varying extents, thereby increasing the entropy of generated sequences.

**Stochastic Decoding.** A separate class of decoding methods sample from the model at generation time, $y_t \sim q(y_t|y_{<t}, x, p_\theta)$. This method has shown to be effective at generating diverse samples and can better explore target design spaces. We consider a top-$k$ sampler (Fan et al., 2018), which restricts sampling to the $k$ most-probable tokens at time-step $t$. This corresponds to restricting sampling to a subset of the vocabulary $U \subset V$. $U$ is the subset of $V$ that maximizes $\sum_{y \in U} p_\theta(y_t|y_{y<t}, x)$:

$$q(y_t|y_{<t}, x, p_\theta) = \begin{cases} \frac{p_\theta(y_t|y_{<t}, x)}{Z} & y_t \in U \\ 0 & \text{otherwise} \end{cases}$$

### 4.2 Recursive Inference

We are given $(x, y) \in (X, Y)$ as a sequence pair where by construction $(x, y)$ has high chemical similarity and $y$ scores higher on a prespecified property compared to $x$. At test time, we are interested in recursively inferring new sequences. Let $y_i$ denote a random sequence for recursive iteration $i$. Let $\{y_i^{(k)}\}_{k=1}^K$ be a set of $K$ outputs generated from $p_\theta(y_i|x)$ when $i = 0$. We use a scoring function $S$ to compute the best of $K$ outputs denoted as $\hat{y}_i$. For $i > 0$, we infer $K$ outputs from $p_\theta(y_i|\hat{y}_{i-1})$. This process is repeated for $n$ iterations.

**Scoring functions.** In principle, all $K$ outputs at iteration $i$ can be independently conditioned on to generate new candidates for iteration $i + 1$. This procedure scales exponentially with respect to space and time as a function of $n$ iterations. Therefore, we introduce a suite of simple ranking strategies to score $K$ output sequences to decide which output becomes the next iteration's source sequence. We consider a likelihood based ranking as well as several chemistry-specific metrics further described in the experiments. Designing well-informed scoring functions can help calibrate the distributional statistics of generated sequences, aid in multi-property optimization, and provide interpretable sequences of iteratively optimal translations.

**Ensemble outputs.** After $n$ recursive iterations, we ensemble the generated outputs $\{y_0, y_1, \ldots, y_n\}_{k=1}^K$ and score the sequences on a desired objective. For property optimization, we return the $\arg\max$. In principle, BBRT is not limited to ensembling recursive outputs from a *single* model. Different modeling choices and molecular representations have different inductive biases, which influence the diversity of generated compounds. BBRT can capitalize on these differences by providing a coherent method to aggregate results.

## 5 Experiments

We apply BBRT to solve unconstrained and multi-property optimization problems. To highlight the generality of our method, we apply recursive translation to both sequence and graph-based translation models. **We show that BBRT generates state-of-the-art results on molecular property optimization using already published modeling approaches**. Next we describe how recursive inference lends itself to interpretability through the generation of molecular traces, allowing practitioners to map the evolution of discovered compounds from known starting points through a sequence of local structural changes. At any point in a molecular trace, users may introduce a "break point" to consider alternative translations thereby personally evaluating the trade-offs between conflicting design objectives. Finally, we apply BBRT to the problem of secondary property optimization by ranking.

**Models.** We apply our inference method to sequence and graph-based molecular representations. For sequences, we use the recently introduced SELFIES molecular representation (Krenn et al., 2019), a sequence-based representation for semantically constrained graphs. Empirically, this method generates a high percentage of valid compounds (Krenn et al., 2019). Using SELFIES, we develop a Seq2Seq model with an encoder-decoder framework. The encoder and decoder are both parameterized by RNNs with Long Short-Term Memory (LSTM) cells (Hochreiter & Schmidhuber, 1997). The encoder is a 2-layer bidirectional RNN and the decoder is a 1-layer unidirectional forward RNN. We also use attention (Bahdanau et al., 2014) for decoding. The hidden representations are non-probabilistic and are optimized to minimize a standard cross-entropy loss with teacher forcing. Decoding is performed using deterministic and stochastic decoding strategies described in

| Method | Penalized logP | | | QED | | |
|---|---|---|---|---|---|---|
| | 1st | 2nd | 3rd | 1st | 2nd | 3rd |
| ZINC-250K | 4.52 | 4.30 | 4.23 | 0.948 | 0.948 | 0.948 |
| ORGAN | 3.63 | 3.49 | 3.44 | 0.896 | 0.824 | 0.820 |
| JT-VAE | 5.30 | 4.93 | 4.49 | 0.925 | 0.911 | 0.910 |
| GCPN | 7.98 | 7.85 | 7.80 | 0.948 | 0.947 | 0.946 |
| JTNN | 5.97 | 4.96 | 4.71 | 0.948 | 0.948 | 0.948 |
| Seq2Seq | 4.65 | 4.53 | 4.49 | 0.948 | 0.948 | 0.948 |
| BBRT-JTNN | **10.13** | **10.10** | **9.91** | 0.948 | 0.948 | 0.948 |
| BBRT-Seq2Seq | 6.74 | 6.47 | 6.42 | 0.948 | 0.948 | 0.948 |

Table 1: Top 3 property scores on penalized logP and QED tasks.

Section 4.1. For graphs, we use a tree-based molecular representation (Jin et al., 2018) with the exact architecture described in Jin et al. (2019b). Decoding is performed using a probabilistic extension with latent variables described in Jin et al. (2019b)—we sample from the prior $k$ times followed by left-to-right greedy decoding.

**Data**. We construct training data by sampling molecular pairs $(X, Y)$ with molecular similarity $sim(X, Y) > \tau$ and property improvement $\delta(Y) > \delta(X)$ for a given property $\delta$. Constructing training pairs with a similarity constraint can help avoid degenerate mappings. In contrast to Jin et al. (2019b), we only enforce the similarity constraint for the construction of training pairs. Similarity constraints are not enforced at test time. Molecular similarity is measured by computing Tanimoto similarity with Morgan fingerprints (Rogers & Hahn, 2010). All models were trained on the open-source ZINC dataset (Irwin et al., 2012). We use the ZINC-250K subset, as described in Gómez-Bombarelli et al. (2018).

**Properties.** For all experiments, we focus on optimizing two well-known drug properties of molecules. First, we optimize the water-octanol partition coefficient (logP). Similar to Jin et al. (2018); Kusner et al. (2017); Gómez-Bombarelli et al. (2018), we consider a penalized logP score that incorporates ring size and synthetic accessibility. The penalized logP score uses a dataset normalization score described in You et al. (2018a). Following Jin et al. (2019b) we extracted 99K translation pairs for training using a similarity constraint of 0.4. Second, we optimize quantitative estimate of drug likeness (QED) (Bickerton et al., 2012). Following Jin et al. (2019b), we construct training pairs where the source compound has a QED score within the range [0.7 0.8] and the target compound has a QED score within the range [0.9 1.0]. While Jin et al. (2019b) evaluates QED performance based on a closed set translation task, we evaluate this model for unconstrained optimization. We extracted a training set of 88K molecule pairs. We report the details on how these properties were computed in Appendix A.

**Scoring functions**. Here we describe scoring functions that are used to rank $K$ outputs for recursive iteration $i$. The highest scoring sequence is used as the next iteration's source compound.

- Penalized logP: Choose the compound with the maximum logP value. This is useful when optimizing logP as a primary or secondary property.

- QED: Choose the compound with the maximum QED value. This is useful when optimizing QED as a primary or secondary property.

- Max Delta Sim: Choose the compound with the highest chemical similarity to the previous iteration's source compound. This is useful for interpretable, molecular traces by creating a sequence of translations with local edits.

- Max Init Sim: Choose the compound with the highest similarity to the initial seed compound. This is useful for input-constrained optimization.

- Min Mol Wt: Choose the compound with the minimum molecular weight. This is useful for rectifying a molecular generation artifact where models optimize logP by simply adding functional groups, thus increasing the molecular weight (Figure 7).

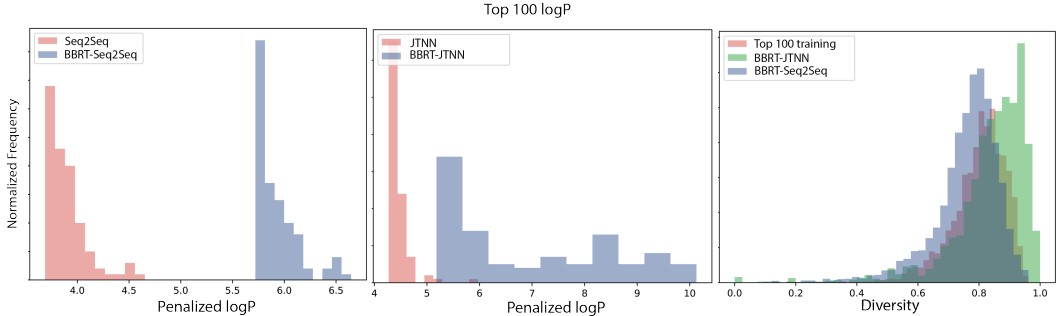

Figure 2: Left and Center: Top 100 logP generated compounds under BBRT-Seq2Seq, BBRT-JTNN, and their non-recursive counterparts. Right: Diversity of top 100 generated compounds under both BBRT models and the top 100 compounds from the training data.

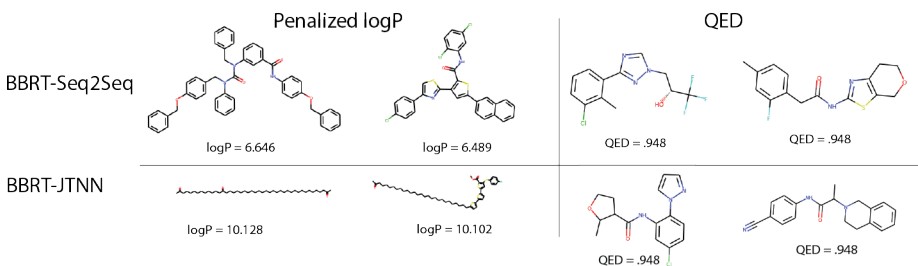

Figure 3: Top scoring compounds for properties logP and QED under BBRT-Seq2Seq and BBRT-JTNN.

Diversity is computed as follows. Let $Y$ be a set of translated compounds. Consistent with the literature (Jin et al., 2019a) we report diversity as

$$\text{DIV}(Y) = \frac{1}{|Y|(|Y| - 1)} \sum_{y \in Y} \sum_{y' \in Y, y'} 1 - \delta(y, y') \tag{2}$$

$\delta$ is the Tanimoto similarity computed on the Morgan fingerprints of $y$ and $y'$.

## 5.1 MOLECULE GENERATION RESULTS

**Setup.** For property optimization, the goal is to generate compounds with the highest possible penalized logP and QED values. For notation, we denote BBRT applied to model X as *'BBRT-X'*. We consider BBRT-JTNN and BBRT-Seq2Seq under 3 decoding strategies and 4 scoring functions. For the logP task, we seed our translations with 900 maximally diverse compounds with an average pairwise diversity of $0.94 \pm 0.04$ relative to $0.86 \pm 0.04$, which is the average pairwise diversity of the training data. Maximally diverse compounds were computed using the MaxMin algorithm (Ashton et al., 2002) on Morgan fingerprints. We found seeding our translations with diverse compounds helped BBRT generate higher scoring compounds (Supplement Fig. 10). For both BBRT applications, we sampled 100 complete sequences from a top-2 and from a top-5 sampler and then aggregated these outputs with a beam search using 20 beams outputting 20 sequences.

**Baselines.** We compare our method with the following state-of-the-art baselines. Junction Tree VAE (JT-VAE; Jin et al. 2018) combines a graph-based representation with latent variables for molecular graph generation. Molecular optimization is performed using Bayesian Optimization on the learned latent space to search for compounds with optimized property scores. JT-VAE has been shown to outperform other methods including CharacterVAE (Gómez-Bombarelli et al., 2018), GrammarVAE (Kusner et al., 2017), and Syntax-Directed-VAE (Dai et al., 2018). We also compare against two reinforcement learning molecular generation methods: Objective-Reinforced Generative Adversarial Networks (ORGAN; Guimaraes et al. 2017) uses SMILES strings (Weininger, 1988), a text-based

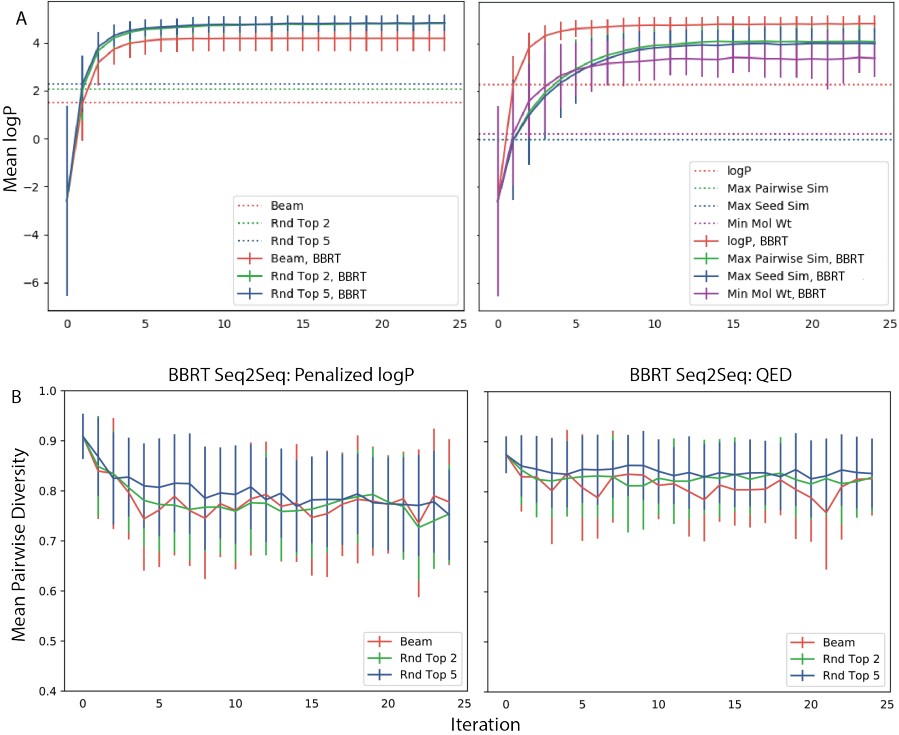

Figure 4: Ablation experiments using BBRT-Seq2Seq. A. Left: Mean logP from 900 translations as a function of recursive iteration for three decoding strategies. Dotted lines denote non-recursive counterparts. Right: Mean logP as a function of recursive iteration for 4 scoring functions. B. Left: Diversity of generated outputs across recursive iterations for logP translation. Right: Diversity of generated outputs across recursive iterations for QED translation.

molecular representation, and the Graph Convolutional Policy Network (GCPN; You et al. 2018a), uses graphs.

We also compare against non-recursive variants of the Seq2Seq and Variational Junction-Tree Encoder-Decoder (JTNN; Jin et al. 2019b) models. Seq2Seq is trained on the SELFIES representation (Krenn et al., 2019). For a fair comparison, we admit similar computational budgets to these baselines. For Seq2Seq we include a deterministic and stochastic decoding baseline. For the deterministic baseline, we use beam search with 20 beams and compute the 20 most probable sequences under the model. For the stochastic baseline, we sample 6600 times from a top-5 sampler. For details on the sampler, we refer readers to Section 4.1. For JTNN, we use the reported GitHub implementation from Jin et al. (2019b). There is not an obvious stochastic and deterministic parallel considering their method is probabilistic. Therefore, we focus on comparing to a fair computational budget by sampling 6600 times from the prior distribution followed by greedy decoding. For fair comparisons to the recursive application, the same corresponding sampling strategies are used, with 100 samples per iteration.

**Results.** Following reporting practices in the literature, we report the top 3 property scores found by each model. Table 1 summarizes these results. The top 3 property scores found in ZINC-250k are included for comparison. For logP optimization, BBRT-JTNN significantly outperforms all baseline models including JTNN, Seq2Seq, and BBRT-Seq2Seq. BBRT-Seq2Seq outperforms Seq2Seq, highlighting the benefits of recursive inference for both molecular representations. For QED property optimization, the two translation models and the BBRT variants all find the same top 3 property scores, which is a new state-of-the-art result for QED optimization.

In Figure 2, we report the top 100 logP compounds generated by both BBRT applications relative to its non-recursive counterparts and observe significant improvements in logP from using BBRT. We also report a diversity measure of the generated candidates for both BBRT models and the top 100 logP compounds in the training data. The JTNN variant produces logP compounds that are more

diverse than the compounds in the training data, while the compounds generated by Seq2Seq are less diverse.

Figure 3 visualizes the top 2 discovered compounds by BBRT-JTNN and BBRT-Seq2Seq under both properties. For logP, while BBRT-JTNN produces compounds with higher property values, BBRT-Seq2Seq's top 2 generated compounds have a richer molecular vocabulary. BBRT-Seq2Seq generates compounds with heterocycles and linkages while BBRT-JTNN generates a chain of linear hydrocarbons, which resemble the top reported compounds in GCPN (You et al., 2018b), an alternative graph-based representation. The stark differences in the vocabulary from the top scoring compounds generated by the sequence- and graph-based representations highlight the importance of using flexible frameworks that can ensemble results across molecular representations.

**Differences between logP and QED.** For logP, BBRT provides a 27% improvement over state-of-the-art for property optimization, while for QED, despite recursive methods outperforming ORGAN, JT-VAE and GCPN, the corresponding non-recursive techniques—Seq2Seq and JTNN baselines perform just as well as with and without BBRT. We argue this might be a result of these models reaching an upper bound for QED values, motivating the importance of the community moving to new metrics in the future (Korovina et al., 2019).

## 5.2 EMPIRICAL PROPERTIES OF RECURSIVE TRANSLATION.

We perform a sequence of ablation experiments to better understand the effect of various BBRT design choices on performance. We highlight the variability in average logP from translated outputs at each iteration with different decoding strategies (Figure 4A left) and scoring functions (Figure 4A right).

**On the importance of stochastic decoding.** For non-recursive and recursive translation models, **stochastic decoding methods outperformed deterministic methods** on average logP scores (Figure 4A left) and average pairwise diversity (Figure 4B) for generated compounds as a function of recursive iteration. Non-greedy search strategies are not common practice in *de novo* molecular design (Gómez-Bombarelli et al., 2018; Kusner et al., 2017; Jin et al., 2019b). While recent work emphasizes novel network architectures and generating diverse compounds using latent variables (Gómez-Bombarelli et al., 2018; Kusner et al., 2017; Jin et al., 2018), we identify an important design choice that typically has been underemphasized. This trend has also been observed in the natural language processing (NLP) literature where researchers have recently highlighted the importance of well-informed search techniques (Kulikov et al., 2018).

Regardless of the decoding strategy, we observed improvements in mean logP with iterations (Figure 2A) when using BBRT. When optimizing for logP, a logP scoring function quickly discovers the best scoring compounds while secondary scoring functions improve logP at a slower rate and do not converge to the same scores (Figure 2A right). This trade-off highlights the role of conflicting molecular design objectives.

For Figure 4A, the standard deviation typically decreased with iteration number $n$. Property values concentrate to a certain range. With property values concentrating, it is reasonable to question whether BBRT produces compounds with less diversity. In Figure 4B we show average pairwise diversity of translated outputs per recursive iteration across 3 decoding strategies and observe decay in diversity for logP. For the best performing decoding strategy, the top-5 sampler, diversity decays from about 0.86 after a single translation to about 0.78 after $n = 25$ recursive translations. This decay may be a product of the data—higher logP values tend to be less diverse than a random set of compounds. For QED (Figure 4B right), we observe limited decay. Differences in decay rate might be attributed to task variability, one being explorative and the other interpolative.

## 5.3 INTERPRETABLE, USER-CENTRIC OPTIMIZATION

For project teams in drug design, the number of generated suggestions can be a practical challenge to prioritization of experiments. We found that interpretable paths of optimization facilitate user adoption.

**Molecular traces.** BBRT generates a sequence of iterative, local edits from an initial compound to an optimized one. We call these optimization paths a *molecular trace* (Figure 5A). We use the Min Delta Sim scoring function to generate traces that have the minimum possible structural changes,

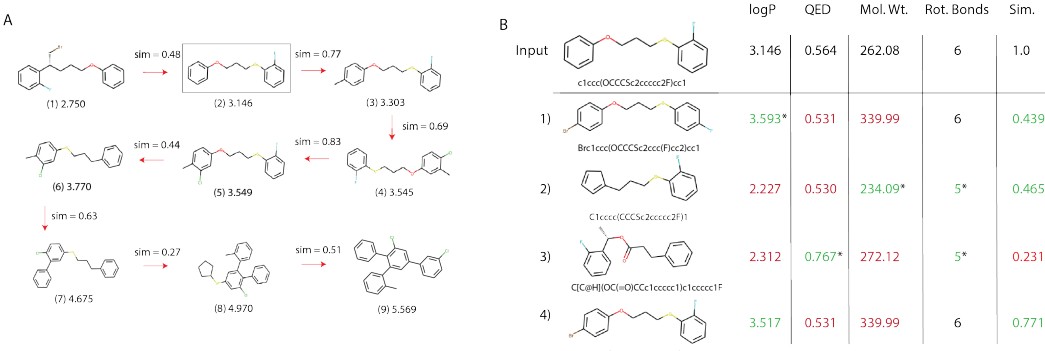

Figure 5: A. Generated molecular trace by ranking intermediate outputs by the maximum pairwise Tanimoto similarity. B. An example molecular breakpoint. Alternative translations are considered from compound (2) each with its own design trade-offs.

while still improving logP. While some graph-based approaches (Jin et al., 2018; You et al., 2018b; Kearnes et al., 2019) can return valid, intermediate graph states that are capable of being interrogated, we liken our molecular traces to a sequence of Free-Wilson analysis (Free & Wilson, 1964) steps towards optimal molecules. Each step represents a local model built using molecular subgraphs with the biological activity of molecules being described by linear summations of activity contributions of specific subgraphs. Consequently, this approach provides interpretability within the chemical space spanned by the subgraphs (Eriksson et al., 2014).

**Molecular breakpoints.** Molecular design requires choosing between conflicting objectives. For example, while increased logP is correlated with poor oral drug-like properties and rapid clearance from the body (Ryckmans et al., 2009), increasing QED might translate to a compound that is structurally dissimilar from the seed compound, which could result in an inactive compound against a target. Our method allows users to "debug" any step of the translation process and consider alternative steps, similar to breakpoints in computer programs. In Figure 5B, we show an example from an BBRT-Seq2Seq model trained to optimize logP. Here, we revisit the translation from step (2) to step (3) in Figure 5A by considering four alternatives picked from 100 stochastically decoded compounds. These alternatives require evaluating the trade-offs between logP, QED, molecular weight, the number of rotational bonds, and chemical similarity with compound (2).

## 5.4 IMPROVING SECONDARY PROPERTIES BY RANKING

We consider secondary property optimization by ranking recursive outputs using a scoring function. This function decides what compound is propagated to the next recursive step. We apply BBRT to Seq2Seq modeling (BBRT-Seq2Seq) and use the trained QED translator described in Section 5.1. The inference task is to optimize QED and logP as the primary and secondary properties respectively. We compare scoring outputs based on QED and logP. In Figure 6A, we compute the average logP per recursive iteration for a set of translated compounds across three decoding strategies. Dotted lines optimize logP as the scoring function while the solid lines optimize QED. For both scoring functions, we report the maximum logP value generated. For all decoding strategies, average logP reaches higher values under scoring by logP relative to scoring by QED. In Figure 6B, we plot average QED values using the same setup and observe that optimizing logP still significantly improves the QED values of generated compounds. This method also discovers the same maximum QED value as scoring by QED. This improvement, however, has trade-offs in the limit for average QED values generated. After $n = 15$ recursive iterations the *average* QED values of the generated compounds under a logP scoring function converge to lower values relative to QED values for compounds scored by QED for all three decoding strategies. We repeat this experiment with JTNN and show similar effects (Figure 8).

Secondary property optimization by ranking extends to variables that are at minimum loosely positively correlated. For QED optimization, the average logP value for unoptimized QED compounds is $0.30\pm1.96$ while for optimized QED compounds the average logP value is $0.79\pm1.45$. Additionally, QED compounds in the target range [0.9 1.0] in the training data had a positive correlation between its logP and QED values (Spearman rank correlation; $\rho = 0.07 \; P < 0.026$). This correlation did

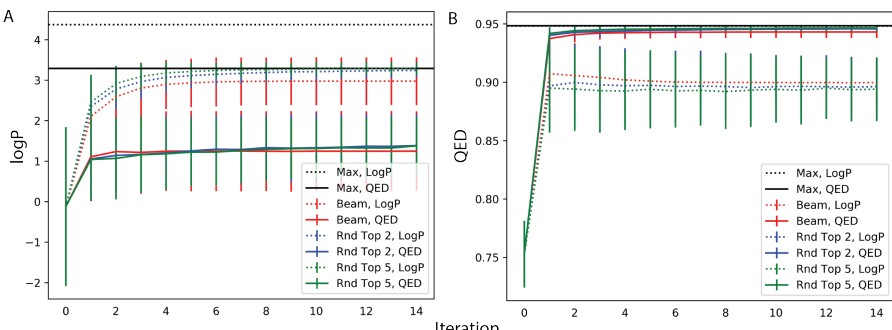

Figure 6: Applying BBRT to multi-property optimization. QED is the primary target and logP is the secondary property. A: Average logP as a function of recursive iteration for three decoding strategies with primary and secondary property scoring functions. B: Average QED as a function of recursive iteration for three decoding strategies with primary and secondary property scoring functions.

not hold for QED compounds in the range [0.7 0.8] unoptimized QED compounds ($\rho = 0.007$, $P < 0.8$).

## 6 FUTURE WORK

We develop BBRT for molecular optimization. BBRT is a simple algorithm that feeds the output of translation models back into the same model for additional optimization. We apply BBRT to well-known models in the literature and produce new state-of-the-art results for property optimization tasks. We describe molecular traces and user-centric optimization with molecular breakpoints. Finally, we show how BBRT can be used for multi-property optimization. For future work, we will extend BBRT to consider multiple translation paths simultaneously. Moreover, as BBRT is limited by the construction of labeled training pairs, we plan to extend translation models to low-resource settings, where property annotations are expensive to collect.

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

## A    RECURSIVE PENALIZED LOGP EXPERIMENTS

**Training details**. For the Seq2Seq model, the hidden state dimension is 500. We use a 2 layer bidirectional RNN encoder and 1 layer unidirectional decoder with attention (Bahdanau et al., 2017). The model was trained using an Adam optimizer for 20 epochs with learning rate 0.001. For the graph-based model, we used the implementation from Jin et al. (2019b), which can be downloaded from: `https://github.com/wengong-jin/iclr19-graph2graph`.

**Property calculation**. Penalized logP is calculated using the implementation from You et al. (2018a). Their implementation utilizes RDKit to compute clogP and synthetic accessibility scores. QED scores are also computed using RDKit.

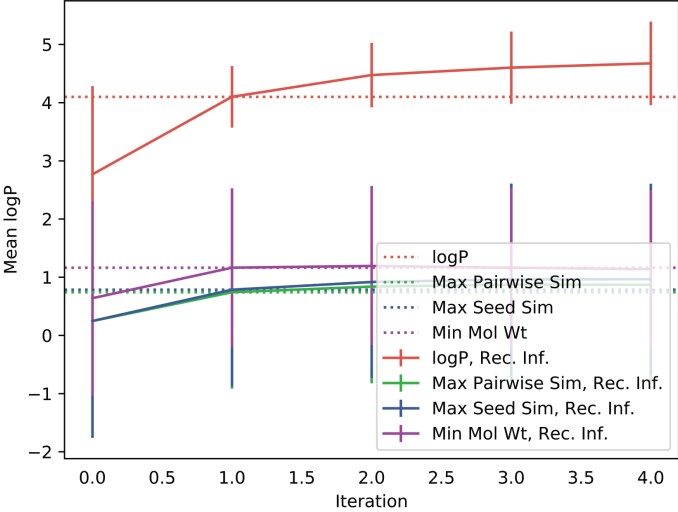

Figure 7: Comparison of scoring functions for BBRT-JTNN. Y-axis is mean logP from 900 translations as a function of recursive iteration. Dotted lines denote non-recursive counterparts. 'Rec. Inf.' is synonymous with BBRT-JTNN.

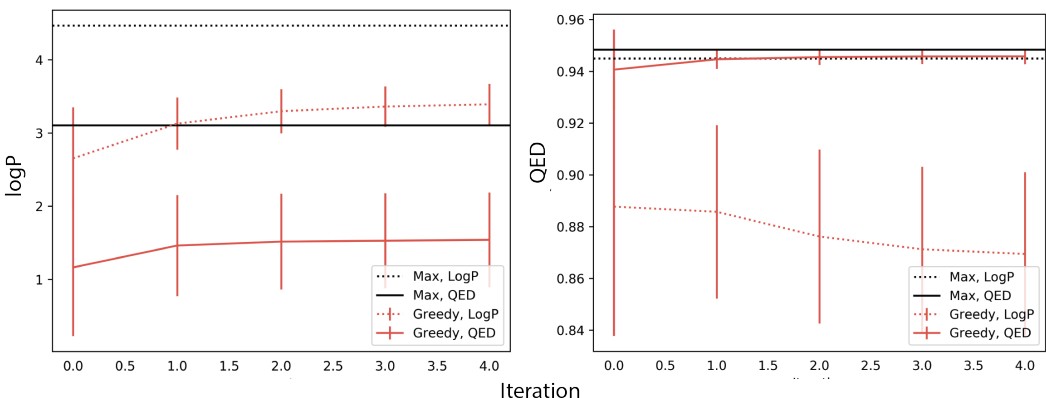

Figure 8: Applying BBRT-JTNN to secondary property optimization. QED is the primary target and logP is the secondary property. Left: Average logP as a function of recursive iteration under two scoring functions—QED and logP. Right: Average QED as a function of recursive iteration for same two scoring functions.

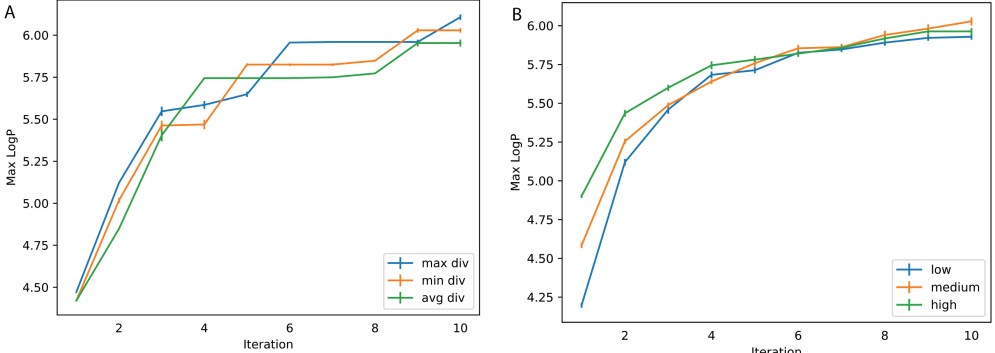

Figure 9: A. Applying BBRT-Seq2Seq to three seed sequence sets with 100 samples each. The MaxMin algorithm was used to select samples with varying levels of diversity (max: 0.94, avg: 0.86, and min: 0.78). For each input sequence, we sampled 100 times from a top5 sampler and ranked samples using logP. Standard error is reported by averaging over 10 BBRT-Seq2Seq runs with different seed sets. B. Applying BBRT-Seq2Seq to three seed sequence sets with 100 samples each and varying logP values (low: logP values $< 1$, medium: values between -1 and 1, high: values $> 1$). Standard error is reported by averaging over 10 runs each with a randomly chosen set of seed sequences.

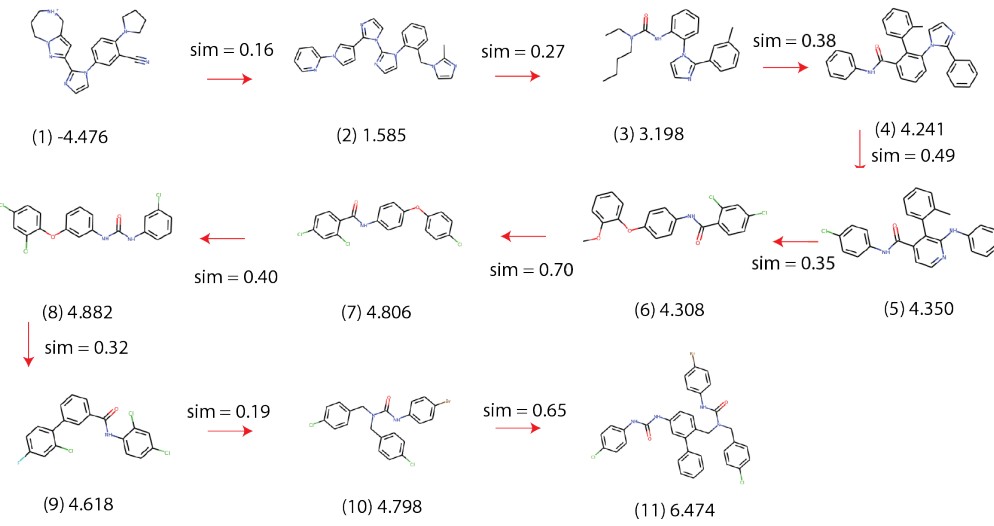

Figure 10: A molecular trace from optimizing logP using a logP scoring function.

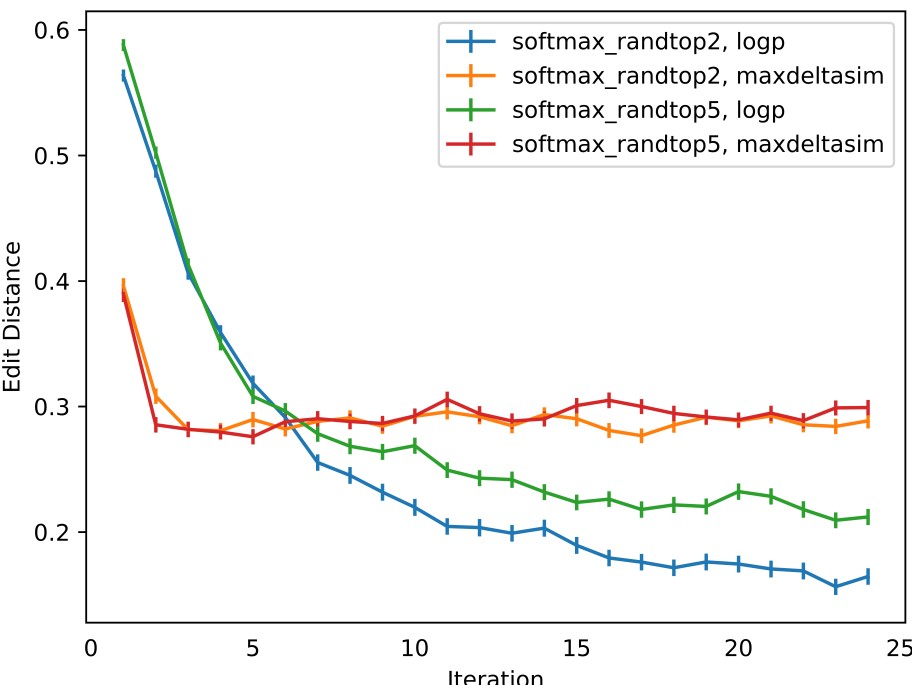

Figure 11: Comparing pairwise Levenshtein edit distances for generated sequences after running BBRT under two different scoring functions (logP and maximum pairwise Tanimoto similarity) and two different decoding strategies (top 2 and top 5 sampling). Standard errors are reported from a population of 900 sequences per iteration.

