# OpenReview forum: "Black Box Recursive Translations for Molecular Optimization"
_ICLR.cc/2020/Conference — Reject_

### Official Review · AnonReviewer1 · 2019-10-12
**Official Blind Review #1**

**Rating:** 6

**Review:**

The authors frame molecule optimization as a sequence-to-sequence problem where a source molecule is translated to a target molecule with improved properties. The authors extend existing methods for improving molecules by applying them recursively over multiple rounds, and show that it is beneficial for optimizing logP but not QED. An advantage over existing methods is that the trajectory of optimized molecules is interpretable. Altogether, I find the paper borderline: it is clearly written but the methodological contribution is incremental, some citations to related work missing, and some parts of the results section are weak. Detailed comments below.

Major comment
============
1. Framing optimizing as a sequence to sequence problem is not new. As described in the related work section, the BBRT is closely related to Jin et al. However, it is not clearly described what the major improvement over Jin et al is. Please clarify ‘their inference method restricts the framework’s application to more general problems.’.  The method is also closely related to Zou et al (https://www.nature.com/articles/s41598-019-47148-x) and Mueller et al (http://proceedings.mlr.press/v70/mueller17a.html), which are not cited in the text. Zou et al used RL to learn to optimize molecules by mutating existing molecules. Mueller et al used Seq2Seq to optimize the sentiment of sentences. Please cite these papers and discuss why BBRT is better.

2. Please compare to ChemBO (http://arxiv.org/abs/1908.01425). The current baselines are one-shot in that they are proposing a batch of molecules once without using the acquired target function label to propose subsequent batches. ChemBO optimizes a target function such as logP over multiple rounds similar to recursive BBRT approach, and should therefore be included as a baseline. Another suitable baseline would be performing BO in the latent space by applying Gomez et Bombarelli recursively (embed molecule; optimize GP in embedding space; decode molecule; iterate).

3. The method names (Graph2Graph, Seq2Seq, R-Graph2Graph, R-Seq2Seq, BBRT-JTNN, …) are not defined in section 5.1-baselines, and used inconsistently. Is JTNN the same as Graph2Graph and does BBRT mean recursive (R-)? This makes is hard to follow the results section.

4. Section 5.1: How does the performance depends on the initial seed of sequences? How sensitive is it i) to the choice of the diversity cutoff, and ii) to the target value of the initial molecules?

5. Fig 4a, right: Is is expected that logP increases fastest when using it as a scoring function. Please show instead QED vs. the number of iterations. QED combines several molecular properties, including logP, and is therefore more suited for quantifying drug likeness.

6. ‘Differences between logP and QED.’ I do not understand this section. Please clarify the goal of an explorative vs interpolative task? Are molecules with the highest QED in the training dataset? Motivate why BBRT does not achieve a higher QED in table 1?

7. ‘The distinction in the vocabulary highlights the usefulness …’. Is the conclusion that representing molecules as sequences is better than representing them as graphs? This would contradict several recent papers on graph-based representations. You are only comparing the top molecules. Is there a significant difference in the complexity between the top 100 (for example) molecules?

8. Section 5.3 is verbose and can shortened to a few sentences saying that applying edits to molecules recursively makes the model interpretable. How do traces look like when logP is used a  selection criteria? How does the trace of the best molecule shown in figure 3 look like? What is the average edit distance between molecules are and intermediate molecules valid? Are transitions plausible?

9. Section 5.4: You are optimizing a single objective (e.g. logP) while reporting in parallel a second objective (e.g. QED). This is not multi-objective optimization, where multiple objectives are optimized in parallel. Optimizing a single objective while reporting a second objective can be also done with methods other than BBRT. Please clarify the take-away message of this paragraph or remove it.


Minor comments
=============
10. Introduction: ‘discrete and unstructured’. Why unstructured? I would say that molecules are structured--they must follow a certain grammar to be valid.

11. Introduction: ‘treating inference as a first class citizen’ is unclear since ‘inference’ is undefined. Either remove this sentence or clarify.

12. Please discuss that BBRT is limited by the need of a labeled dataset for constructing training pairs.

13. Section 5.1, ‘Similar computational budget’. How did you quantify the computation budget?

14. Section 5.1, ‘In Fig 2, we report’. Do you mean Fig 3? Same as with ‘Fig 3’ in the following paragraph.

**Experience Assessment:**

I have published one or two papers in this area.

**Review Assessment: Checking Correctness Of Derivations And Theory:**

N/A

**Review Assessment: Checking Correctness Of Experiments:**

I carefully checked the experiments.

**Review Assessment: Thoroughness In Paper Reading:**

I read the paper thoroughly.

---

> ### Author Response · Authors · 2019-11-15
> **Response Part 1**
>
> We thank the reviewer for their thoughtful response.
>
> > "1. Framing optimizing as a sequence to sequence problem is not new. As described in the related work section, the BBRT is closely related to Jin et al. However, it is not clearly described what the major improvement over Jin et al is. Please clarify ‘their inference method restricts the framework’s application to more general problems.’.  The method is also closely related to Zou et al (https://www.nature.com/articles/s41598-019-47148-x) and Mueller et al (http://proceedings.mlr.press/v70/mueller17a.html), which are not cited in the text. Zou et al used RL to learn to optimize molecules by mutating existing molecules. Mueller et al used Seq2Seq to optimize the sentiment of sentences. Please cite these papers and discuss why BBRT is better."
>
> Jin et al. 2019 translate source graphs to improved target graphs while retaining high structural similarity. BBRT extends the translation framework from similarity-constrained optimization to the more general unconstrained setting of finding the best scoring molecules regardless of its similarity to a seed compound. Our results show that BBRT provides significant improvements in unconstrained molecular optimization relative to its non-recursive peer--just decoding from a translation model (Jin et al. 2019).
>
> Thank you for these references. Zhou et al. 2019 is cited in the text--in the first paragraph of Section 2 Related Work: “Molecular optimization has been approached with reinforcement learning (Popova et al., 2018; Zhou et al., 2019).”
>
> Mueller et al use a variational autoencoder coupled with constrained optimization in the latent space to decode better sequences. There are two differences: 1) Mueller et al. learns a marginal density p(x) while we directly improve on compounds by repeatedly applying a model that learns p(y|x) and 2) they focus on similarity-constrained optimization (optimize the sentiment of sentences without changing the sentence too much) while we focus on finding the best sequences more generally. Zhou et al. perform molecular optimization with reinforcement learning. While reinforcement learning techniques have shown great promise for molecular optimization, we focus on a different problem--one of molecular optimization via direct supervised translations. We argue that framing translation as optimization allows us to solve an easier supervised learning problem, which we have shown can lead to improved results. While we do not compare against Zhou et al. 2019 directly, we do provide comparisons to two other reinforcement learning techniques (ORGAN and GCPN).
>
> > "2. Please compare to ChemBO (http://arxiv.org/abs/1908.01425). The current baselines are one-shot in that they are proposing a batch of molecules once without using the acquired target function label to propose subsequent batches. ChemBO optimizes a target function such as logP over multiple rounds similar to recursive BBRT approach, and should therefore be included as a baseline. Another suitable baseline would be performing BO in the latent space by applying Gomez et Bombarelli recursively (embed molecule; optimize GP in embedding space; decode molecule; iterate)."
>
> We tried downloading ChemBO (https://github.com/ks-korovina/chembo), but observed errors when running the program. Given the short timeline of rebuttals, we were unable to compare to this method, which we will leave for future work. We note, however, that 3 of the 5 baselines (ORGAN, JT-VAE, GCPN) are not “one-shot”, they are iterative methods. ORGAN and GCPN optimize molecules with reinforcement learning--iteratively generating molecules, computing a reward, and repeating. JT-VAE performs Bayesian Optimization in a learned latent space to find better compounds. This is an identical set up to ChemBO in terms of the optimization--the main difference between the two being the molecular representation used (learned latent embeddings versus a kernel on molecular graphs). We do not compare directly to Char-VAE (Gómez-Bombarelli et al. 2018) because JT-VAE is in the same class of methods (VAE plus post-hoc BO in latent space), but JT-VAE has been shown empirically to find better compounds (Jin et al. 2018). We appreciate your suggestion to apply Char-VAE recursively. Optimizing molecules in an embedding space using Bayesian Optimization will find a local minima, so we have no reason to believe that recursive applications of this model will improve the compounds discovered from the initial (fully optimized) application of BO. It’s possible that recursive applications can find different local minima, but random restarts can also deal with this issue. For our application, recursive applications of a translation model makes sense because a single translation does not correspond to a local minima. In the BO case, it does.

---

> ### Author Response · Authors · 2019-11-15
> **Response Part 2**
>
> > "3. The method names (Graph2Graph, Seq2Seq, R-Graph2Graph, R-Seq2Seq, BBRT-JTNN, …) are not defined in section 5.1-baselines, and used inconsistently. Is JTNN the same as Graph2Graph and does BBRT mean recursive (R-)? This makes is hard to follow the results section."
>
> We have fixed the notation.
>
> > "4. Section 5.1: How does the performance depends on the initial seed of sequences? How sensitive is it i) to the choice of the diversity cutoff, and ii) to the target value of the initial molecules?"
>
> Please see Figure 9 in the supplement. We have added experiments to better understand how performance varies depending on the diversity and the property values of the seed compounds. In Fig. 9A, we selected 3 sets of 100 seeds with different diversity levels (min, avg, max) computed with the MinMax algorithm (described in Section 5.1 setup), and applied BBRT to these three sets. Standard error is reported after running this experiment 10 times with different sets of seed compounds chosen each time (using the randomness in MinMax). We observe improvements in performance using a set of seed sequences with max diversity.
>
> For seed molecules with high property values, there are typically no “target values” available in the training data. Instead, in Fig. 9B, we assessed how performance depends on the property values of the initial molecules themselves. We observed improved performance in early iterations using seed molecules with high property values, although in later iterations, the seed molecule property values does not seem to make much of a difference.
>
> > "5. Fig 4a, right: Is is expected that logP increases fastest when using it as a scoring function. Please show instead QED vs. the number of iterations. QED combines several molecular properties, including logP, and is therefore more suited for quantifying drug likeness."
>
> Fig. 4A right highlights the performance tradeoffs when using different scoring functions, which can allow users to weigh between competing tradeoffs (e.g. interpretability vs performance). The fact that using logP as a scoring function when optimizing for logP finds the best logP compounds is useful information if the end goal of the user is to find the highest logP scoring molecules. In Fig. 6, we show the optimization of QED while using logP as a scoring function (the reverse of your suggestion). When optimizing for QED, using logP as a scoring function generates compounds that score highly on both metrics.
>
> > "6. ‘Differences between logP and QED.’ I do not understand this section. Please clarify the goal of an explorative vs interpolative task? Are molecules with the highest QED in the training dataset? Motivate why BBRT does not achieve a higher QED in table 1?"
>
> This section has been clarified and the “explorative vs interpolative” language has been removed. Despite our BBRT method outperforming ORGAN, JT-VAE and GCPN on the QED task, we observe the corresponding non-recursive techniques---Seq2Seq and JTNN baselines performed just as well without recursive inference. This might be a result of the translation models finding compounds that have reached a max ceiling for the best scoring QED compounds (we haven’t seen any paper report higher QED values than 0.948). Additionally, molecules with the highest QED values are in the training dataset. We leave for future work the application of BBRT to other metrics and experimental designs that leave out the best scoring compounds from the training data.
>
> > "7. ‘The distinction in the vocabulary highlights the usefulness …’. Is the conclusion that representing molecules as sequences is better than representing them as graphs? This would contradict several recent papers on graph-based representations. You are only comparing the top molecules."
>
> We do not make any claim that a sequence-based representation is better than a graph-based representation. Instead we argue that sequence and graph-based representations are complementary approaches that empirically leverage different vocabularies for top scoring compounds. Flexible frameworks (like BBRT) that enable the ensembling of results across molecular representations is key to diverse generation. For future work, we would like to include additional molecular representations (including grammars and alternative graph-based representations).
>
> > "Is there a significant difference in the complexity between the top 100 (for example) molecules?"
>
> Please see Fig. 2 right. We report the diversity of the top 100 generated compounds under BBRT-JTNN, BBRT-Seq2Seq and the top 100 compounds from the training data.

---

> ### Author Response · Authors · 2019-11-15
> **Response Part 3**
>
> > "8. Section 5.3 is verbose and can shortened to a few sentences saying that applying edits to molecules recursively makes the model interpretable. How do traces look like when logP is used a  selection criteria? How does the trace of the best molecule shown in figure 3 look like?"
>
> We appreciate your feedback regarding concision. Section 5.3 has been shortened. Please see Fig. 10 in the supplement. We have added a new molecular trace for a high scoring logP compound (comparable to the best molecule shown in Fig. 3) using logP as the scoring function.
>
> > "What is the average edit distance between molecules"
>
> We have added Fig. 11 to the supplement, which reports the average Levenshtein edit distance between molecules under two decoding strategies and two scoring functions.
>
>  > "are and intermediate molecules valid? Are transitions plausible?"
>
> Following the results reported in the SELFIES paper (Krenn et al. 2019), most generated molecules (including intermediate molecules) are valid. Empirically we observed greater than 99.9% validity. The transitions appear to be plausible. We report two molecular traces in the paper. Additionally, the edit distance plot (Fig. 11) shows a tradeoff between edit distance between pairwise steps and decoding strategy. For softmax sampling from top 2 most likely at each step, the edit distance is highly consistent, while for top 5 sampling, the edit distance is big initially but then it sharply drops off.
>
> > "9. Section 5.4: You are optimizing a single objective (e.g. logP) while reporting in parallel a second objective (e.g. QED). This is not multi-objective optimization, where multiple objectives are optimized in parallel. Optimizing a single objective while reporting a second objective can be also done with methods other than BBRT. Please clarify the take-away message of this paragraph or remove it."
>
> The primary translation model is indeed optimizing a single objective. We describe how BBRT enables scoring and ranking intermediate outputs by a second property, thus propagating molecules that score highly on the second property (relative to other sampled outputs at each step) forward. “Ranking by a secondary property” is a different computation than just reporting a second objective and can be viewed as a form of optimization (albeit a simple one). In Fig. 6A left, the solid lines denote reporting logP (the second objective) while optimizing and scoring by QED. The dotted lines report logP while optimizing QED and scoring by logP. We observe a significant improvements in logP when scoring by logP as opposed to merely reporting its value.  This shows that scoring by the second objective can improve that property’s values without explicit joint optimization.
>
> We understand multi-objective optimization is an overloaded term that can be confusing here given our actual procedure. We have changed the section title to “Improving secondary properties by ranking”.
>
>
> > "Minor comments
> =============
> 10. Introduction: ‘discrete and unstructured’. Why unstructured? I would say that molecules are structured--they must follow a certain grammar to be valid."
>
> While molecules themselves are highly structured objects, chemical space is unstructured (e.g. there is no canonical ordering of all chemicals).
>
> > "11. Introduction: ‘treating inference as a first class citizen’ is unclear since ‘inference’ is undefined. Either remove this sentence or clarify."
>
> We have clarified “inference” with “decoding strategy”.
>
> > "12. Please discuss that BBRT is limited by the need of a labeled dataset for constructing training pairs."
>
> This has been added to the paper.
>
> > "13. Section 5.1, ‘Similar computational budget’. How did you quantify the computation budget?"
>
> For a fair comparison to the non-recursive peers (Seq2Seq and JTNN), we wanted to make sure we decoded the same number of samples as the aggregate number of decoded samples across a BBRT experiment (which includes using 3 decoding strategies and 4 scoring functions). We simply computes number of decoded samples (100) * 3 decoding strategies * 4 scoring functions * number of total recursive iterations to arrive at the “computational budget”. For a fair comparison in terms of model capacity, we used a hidden state dimension of 500, which seemed comparable to the JTNN model baseline.
>
>
> > "14. Section 5.1, ‘In Fig 2, we report’. Do you mean Fig 3? Same as with ‘Fig 3’ in the following paragraph."
>
> Figure 2 reports the top 100 logP generated compounds using BBRT (and its non-recursive peer) for two molecular representation. Figure 3 reports the top 2 compounds under BBRT-Seq2Seq and BBRT-JTNN. This appears to be correct in the text.

---

### Official Review · AnonReviewer3 · 2019-10-22
**Official Blind Review #3**

**Rating:** 3

**Review:**


The paper builds on existing translation models developed for molecular optimization, making an iterative use of sequence to sequence or graph to graph translation models by wrapping them in a meta-procedure. The primary contribution is really just to apply the translation models iteratively, i.e., feeding translation outputs from the models back in as inputs for retranslation. A few strategies are introduced to score / rank candidates before they are chosen for retranslation. The overall idea is very simple, and is likely to work in some basic cases where the property has a natural "additive" nature, e.g., logP that you can improve by adding functional groups. This is recognized but not really controlled in the paper except for selecting for input similarity before retranslating. Moreover, I don't think that you really ever want to just maximize logP for any drug so this particular task is a bit artificial in the first place. Other properties are not additive in the same sense, e.g., drug likeness or QED, and the method doesn't appear to improve it (though, to be fair, there may be a ceiling effect for QED in particular).

One of the main ways that one can control the final output in the iterated translation process is by judiciously selecting or ranking candidates for retranslation. The authors use essentially the score from the model itself, similarity to input, and some basic chemistry metrics to do that. Wouldn't it be much better to train a separate ranking method to guide the iterative steps?

The empirical results are clean though not convincing (see the logP discussion above). Additional properties should be included to demonstrate that the method might actually have some practical value, i.e., generalize beyond additive logP. Multi-property optimization would be one possible setting since de novo models have a hard time to reach intersections of different property constraints. Abstractly, one could imagine that an iterative, successively guided approach could work well. The proposed approach in the paper is somewhat undeveloped. It merely uses a translation model for the primary property, and ranks candidates by the other. This is unlikely to get you to any challenging intersections. Also, since logP was always one of the properties effectiveness in this regard is not really demonstrated either. A slightly more sophisticated approach might use relaxed, separately trained ranking models in intermediate steps, successively tightened towards the intersection as the iteration progresses. E.g.,

Brookes et al., Design by adaptive sampling, arXiv:1810.03714

The paper is clearly written but for such a simple method one would need really convincing results and experiments. Maybe better as a workshop submission?

**Experience Assessment:**

I have published in this field for several years.

**Review Assessment: Checking Correctness Of Derivations And Theory:**

I carefully checked the derivations and theory.

**Review Assessment: Checking Correctness Of Experiments:**

I assessed the sensibility of the experiments.

**Review Assessment: Thoroughness In Paper Reading:**

I read the paper thoroughly.

---

> ### Author Response · Authors · 2019-11-15
> **Response Part 1**
>
> We thank the reviewer for their thoughtful response.
>
> > "The paper builds on existing translation models developed for molecular optimization, making an iterative use of sequence to sequence or graph to graph translation models by wrapping them in a meta-procedure. The primary contribution is really just to apply the translation models iteratively, i.e., feeding translation outputs from the models back in as inputs for retranslation. A few strategies are introduced to score / rank candidates before they are chosen for retranslation. The overall idea is very simple, and is likely to work in some basic cases where the property has a natural "additive" nature, e.g., logP that you can improve by adding functional groups."
>
> We use penalized logP, which penalizes simple strategies like adding functional groups. We agree, however, this is still an artifact of the property that can’t be fully fixed.
>
> > "This is recognized but not really controlled in the paper except for selecting for input similarity before retranslating."
>
> We introduced a scoring function that ranks by the minimum molecular weight. We use this scoring function to help control for the additive nature of logP and observe that ranking by the minimum molecular weight can find better compounds than its non-recursive peer when ranking by minimum molecular weight and non-recursive Seq2Seq scoring by logP.
>
> > "The empirical results are clean though not convincing (see the logP discussion above). Additional properties should be included to demonstrate that the method might actually have some practical value, i.e., generalize beyond additive logP. "
>
> We leave for future work the optimization of properties beyond logP and QED.
>
> > "Multi-property optimization would be one possible setting since de novo models have a hard time to reach intersections of different property constraints. Abstractly, one could imagine that an iterative, successively guided approach could work well. The proposed approach in the paper is somewhat undeveloped. It merely uses a translation model for the primary property, and ranks candidates by the other. This is unlikely to get you to any challenging intersections. Also, since logP was always one of the properties effectiveness in this regard is not really demonstrated either. A slightly more sophisticated approach might use relaxed, separately trained ranking models in intermediate steps, successively tightened towards the intersection as the iteration progresses. E.g., Brookes et al., Design by adaptive sampling, arXiv:1810.03714"
>
> We agree that an iterative, successively guided approach to multi-property optimization is an interesting direction. While our approach of translating according to a primary property and propagating molecules that score highly on a secondary property does not jointly optimize both properties, we have found this simple approach to work quite well. In Fig. 6A, regardless of the decoding strategy used (beam or stochastic decoding), ranking by a secondary property always produces improves in the mean logP of the generated candidates relative to ranking by the primary property. More importantly, the primary properties values do not degrade that significantly as evident by Fig. 6B (they are still very high relative to the seed sequences ~0.90 vs ~0.75).

---

### Official Review · AnonReviewer4 · 2019-10-30
**Official Blind Review #4**

**Rating:** 3

**Review:**

This paper presents a translation-based method for molecular property optimization. It uses a sequence- or graph-based encoder/decoder framework to produce molecules with (hopefully) improved properties, then feeds a subset of these molecules back into the encoder/decoder to generate a new set of molecules; this process is repeated for a fixed number of iterations to arrive at a final set of "optimized" molecules. The method is agnostic to the form of the encoder/decoder; the emphasis is on the iterative approach. Additionally, this approach enables visualization of "molecular" traces that can reveal pathways between molecules that follow relationships similar to matched molecular pairs. The work extends related work in translation-based property optimization [6]. The paper is well-written and generally easy to read.

The method is evaluated on two tasks, logP and QED. Both of these are computed properties that have known issues (see the discussion in [3]), but I understand that these properties are used in many publications and are thus easy to compare. The method presented here performs similar to others on a QED task. They claim superior results on the logP task, but I have concerns about the fairness of the comparison since logP can be exploited by very simple models if there are no limits on the size of the generated molecules (or, similarly, the number of tokens/generative steps allowed for each molecule). Additionally, the authors claim to perform multi-objective optimization but do not actually do this.

The iterative nature of this method is very interesting. However, my concerns about the types of experiments and comparisons that were done (see below for more details) are big enough that I cannot approve this paper in its current form. Weak reject.

Specific notes (starting with page number):

- 1: "potential druggable candidates" does not make sense; compounds are not "druggable" (their targets are), although they may be "drug-like".
- 2: Consider citing Kramer et al.'s seminal work on matched molecular pairs [1].
- 3: Please explain what it means for y to "paraphrase" x?
- 5: For your logP experiments, you need to be more clear about how you are comparing to other models. You are guaranteed to get to higher logP values if you can generate larger molecules (more tokens) than the baselines, since logP is essentially linear in the number of carbons. Are you doing something to limit the number of tokens you can generate in each iteration? Or why should I believe these comparisons are fair?
- 5: In Table 1, note that some literature uses a "normalized" penalized logP, while others use the formula directly without a dataset-specific normalization (which can appear to give better results). Can you confirm which you are using here and whether the baseline models are the same?
- 5: The results in Table 1 would be more compelling if they were not divorced from their starting points. Please include information about the similarity of these molecules to the starting molecule as well as the property delta. Consider an approach like Jin et. al [2], where results were specifically categorized by similarity constraints.
- 5: "All models were trained on the open-source ZINC dataset." What subset of ZINC are you using?
- 5: The supplementary figure showing that logP is broken is missing?
- 5: "Consistent with the literature we report diversity as...". Please cite some literature that you are consistent with.
- 6: "we sample 100 times from a top-2..."; does this mean you are doing 100 iterations? Sampling 100 times from the same top-2 sampler doesn't really make sense, but I'm not entirely sure what you are describing here.
- 7: Figure 4 says these are "ablation" experiments. What exactly are you ablating?
- 8: You state that better performance on logP and similar performance on QED is not known in the literature. In fact, the MolDQN paper [3] calls this out explicitly (and also contains a discussion of bounded vs. unbounded logP).
- 8: "Recent RL methods focus on molecular construction and are therefore not well-suited for the generation of molecular traces"; I disagree with this. RL methods that can start from a predefined graph have the ability to move between compounds, possibly in a way that is orthogonal to traditional similarity-based exploration (see the discussion of "MDP edit distance" in [4]). Also note that one of the key features of graph-based generators like [2] and [5] is that all of the intermediate states are valid, so you could do similar molecular traces for interpretability (although your differences are more like MMPs with functional group-level deltas).
- 9: "synthetic chemists can carry out the individual steps of a molecular trace..."; in general this is not true. The known medicinal chemistry transformations are a relatively small set of operations, and your molecular traces are unlikely to capture them in any systematic way. Please avoid making claims like this unless you can back them up with experimental evidence or comparisons to models explicitly trained for synthetic route planning.
- 9: These experiments are not multi-property optimization. Measuring the value of a secondary property while optimizing a primary property is not the same as optimizing them both simultaneously. The latter requires some strategy for incorporating both property values into the decision function, such as scalarizing (see "multi-objective optimization" on Wikipedia).

References:

[1] Kramer, C. et al. Learning Medicinal Chemistry Absorption, Distribution, Metabolism, Excretion, and Toxicity (ADMET) Rules from Cross-Company Matched Molecular Pairs Analysis (MMPA). J. Med. Chem. 61, 3277–3292 (2018).
[2] Jin, W., Barzilay, R. & Jaakkola, T. Junction Tree Variational Autoencoder for Molecular Graph Generation. arXiv [cs.LG] (2018).
[3] Zhou, Z., Kearnes, S., Li, L., Zare, R. N. & Riley, P. Optimization of Molecules via Deep Reinforcement Learning. Sci. Rep. 9, 10752 (2019).
[4] Kearnes, S., Li, L. & Riley, P. Decoding Molecular Graph Embeddings with Reinforcement Learning. arXiv [cs.LG] (2019).
[5] You, J., Liu, B., Ying, R., Pande, V. & Leskovec, J. Graph Convolutional Policy Network for Goal-Directed Molecular Graph Generation. arXiv [cs.LG] (2018).
[6] Jin, W., Yang, K., Barzilay, R. & Jaakkola, T. Learning Multimodal Graph-to-Graph Translation for Molecular Optimization. arXiv [cs.LG] (2018).

**Experience Assessment:**

I have published one or two papers in this area.

**Review Assessment: Checking Correctness Of Derivations And Theory:**

I did not assess the derivations or theory.

**Review Assessment: Checking Correctness Of Experiments:**

I assessed the sensibility of the experiments.

**Review Assessment: Thoroughness In Paper Reading:**

I made a quick assessment of this paper.

---

> ### Author Response · Authors · 2019-11-15
> **Response Part 1**
>
> We thank the reviewer for their thoughtful response.
>
> > "Review: This paper presents a translation-based method for molecular property optimization. It uses a sequence- or graph-based encoder/decoder framework to produce molecules with (hopefully) improved properties, then feeds a subset of these molecules back into the encoder/decoder to generate a new set of molecules; this process is repeated for a fixed number of iterations to arrive at a final set of "optimized" molecules. The method is agnostic to the form of the encoder/decoder; the emphasis is on the iterative approach. Additionally, this approach enables visualization of "molecular" traces that can reveal pathways between molecules that follow relationships similar to matched molecular pairs. The work extends related work in translation-based property optimization [6]. The paper is well-written and generally easy to read.
>
> The method is evaluated on two tasks, logP and QED. Both of these are computed properties that have known issues (see the discussion in [3]), but I understand that these properties are used in many publications and are thus easy to compare. "
>
> We agree there are problems with logP and QED, but we leave the optimization of new properties for future work.
>
> >The method presented here performs similar to others on a QED task.
>
> They claim superior results on the logP task, but I have concerns about the fairness of the comparison since logP can be exploited by very simple models if there are no limits on the size of the generated molecules (or, similarly, the number of tokens/generative steps allowed for each molecule)."
>
> We use penalized logP, which takes into account ring size and synthetic accessibility. In principle, this property should not reward “daisy chaining” carbon bonds. The penalization, however, is not perfect and is certainly a limitation of using this property.
>
> > "Additionally, the authors claim to perform multi-objective optimization but do not actually do this."
>
> We describe how BBRT enables scoring and ranking intermediate outputs by a second property, thus propagating molecules that score highly on the second property (relative to other sampled outputs at each step) forward. “Ranking by a secondary property” is a different computation than just reporting a second objective and can be viewed as a form of optimization (albeit a simple one). In Fig. 6A left, the solid lines denote reporting logP (the second objective) while optimizing and scoring by QED. The dotted lines report logP while optimizing QED and scoring by logP. We observe a significant improvements in logP when scoring by logP as opposed to merely reporting its value.  This shows that scoring by the second objective can improve that property’s values without explicit joint optimization.
>
> We understand multi-objective optimization is an overloaded term that can be confusing here given our actual procedure. We have changed the section title to “Improving secondary properties by ranking”.
>
> > "- 1: "potential druggable candidates" does not make sense; compounds are not "druggable" (their targets are), although they may be "drug-like"."
>
> Thank you for this correction.
>
> > "- 2: Consider citing Kramer et al.'s seminal work on matched molecular pairs [1]."
>
> We have added this citation.
>
> > "- 3: Please explain what it means for y to "paraphrase" x?"
>
> This language has been removed and the setup has been clarified.
>
> > "- 5: For your logP experiments, you need to be more clear about how you are comparing to other models. You are guaranteed to get to higher logP values if you can generate larger molecules (more tokens) than the baselines, since logP is essentially linear in the number of carbons. Are you doing something to limit the number of tokens you can generate in each iteration? Or why should I believe these comparisons are fair?"
>
> As mentioned above, penalized logP is not linear in the number of carbons. We compute penalized logP in an identical manner to JTNN (Jin et al. 2019), which uses the same setup as GCPN (You et al. 2018). Additionally, in Fig. 4A right we show results where we score by minimum molecular weight to explicitly counteract this logP artifact by choosing compounds that minimize the number of added functional groups while still improving on penalized logP.
>
> > - 5: In Table 1, note that some literature uses a "normalized" penalized logP, while others use the formula directly without a dataset-specific normalization (which can appear to give better results). Can you confirm which you are using here and whether the baseline models are the same?"
>
> We use the normalized penalized logP measure as described in Jin et al. 2019 and You et al. 2018.

---

> ### Author Response · Authors · 2019-11-15
> **Response Part 2**
>
> > "- 5: The results in Table 1 would be more compelling if they were not divorced from their starting points. Please include information about the similarity of these molecules to the starting molecule as well as the property delta. Consider an approach like Jin et. al [2], where results were specifically categorized by similarity constraints."
>
> BBRT extends the translation set up from similarity constrained optimization to unconstrained molecular optimization. The best compounds reported in Table 1 typically have low similarity to the input compound (by design) but are highly similar to the compound from the previous recursive step. We present two molecular traces to highlight the evolution of any compound from its seed to the final compound (Fig. 5A and Fig. 10).
>
> > "- 5: "All models were trained on the open-source ZINC dataset." What subset of ZINC are you using?"
>
> We use the 250K subset from Kusner et al. 2017.
>
> > "- 5: "Consistent with the literature we report diversity as...". Please cite some literature that you are consistent with."
>
> We have added a reference.
>
> > "- 6: "we sample 100 times from a top-2..."; does this mean you are doing 100 iterations? Sampling 100 times from the same top-2 sampler doesn't really make sense, but I'm not entirely sure what you are describing here."
>
> We have clarified this sentence. “For both BBRT applications, we sampled 100 complete sequences from a top-2 and from a top-5 sampler and then aggregated these outputs with a beam search using 20 beams and outputting 20 compounds”.
>
> A top-2 sampler samples from the top 2 most likely tokens at each step of decoding until we hit a stop token. We sample 100 complete sequences following this process.
>
> > "- 7: Figure 4 says these are "ablation" experiments. What exactly are you ablating?"
>
> Table 1 presents the top scoring compounds from aggregating across decoding strategies (deterministic and stochastic decoding) and scoring functions (logP, max pairwise sim, max seed sim, min mol wt.). Fig. 4 disentangles these design decisions and shows how each of these individual components impact performance.
>
> > "- 8: You state that better performance on logP and similar performance on QED is not known in the literature. In fact, the MolDQN paper [3] calls this out explicitly (and also contains a discussion of bounded vs. unbounded logP)."
>
> We were simply stating that using a translation model (whether its graph or sequence based) coupled with stochastic decoding provides better performance on logP (relative to just using deterministic decoding) and provides state-of-the-art results on QED. To the best of our knowledge, this was not known before.
>
> > "- 8: "Recent RL methods focus on molecular construction and are therefore not well-suited for the generation of molecular traces"; I disagree with this. RL methods that can start from a predefined graph have the ability to move between compounds, possibly in a way that is orthogonal to traditional similarity-based exploration (see the discussion of "MDP edit distance" in [4]). Also note that one of the key features of graph-based generators like [2] and [5] is that all of the intermediate states are valid, so you could do similar molecular traces for interpretability (although your differences are more like MMPs with functional group-level deltas)."
>
> Thank you for these references. We have modified the text to mention that there are alternative methods for generating molecular traces using graph-based methods.

---

> ### Author Response · Authors · 2019-11-15
> **Response Part 3**
>
> > "- 9: "synthetic chemists can carry out the individual steps of a molecular trace..."; in general this is not true. The known medicinal chemistry transformations are a relatively small set of operations, and your molecular traces are unlikely to capture them in any systematic way. Please avoid making claims like this unless you can back them up with experimental evidence or comparisons to models explicitly trained for synthetic route planning."
>
> Our claim is not that molecular traces correspond to synthetic routes. Just as there is an association between chemical structure and activity (and minor perturbations to the structure should not change the activity by too much), there is an association between chemical structures and the synthetic routes to make those structures. If pairwise compounds in a molecular trace are similar, their corresponding synthetic routes will be similar, aiding in the efficiency of chemical synthesis (if you can make one step i, it should follow that you can make step i+1).
>
> We have removed this discussion from the paper as we have concluded that it is only tangentially related.
>
> > "- 9: These experiments are not multi-property optimization. Measuring the value of a secondary property while optimizing a primary property is not the same as optimizing them both simultaneously. The latter requires some strategy for incorporating both property values into the decision function, such as scalarizing (see "multi-objective optimization" on Wikipedia)."
>
> Please see our comment in "Response Part 1".

---

### Official Review · AnonReviewer2 · 2019-10-31
**Official Blind Review #2**

**Rating:** 6

**Review:**

This paper presents a novel approach to generating molecules using Black Box Recurrent Translation.
The authors uses existing machine-translation inspired schemes to generate new, similar, molecules with better properties according to some measure.
Then, the recursion takes the top K best molecules and runs another iteration to generate even better molecules, ad infinitum.
The authors use the newly introduced SELFIES strings as vocabulary for generation.
The authors also analyze the decoding strategy, and how the process generates interpretable molecular traces.
Relating to wetlab work, having a molecular trace available from the recursive translation scheme is valuable for drug-sythesis.
The authors also show that this technique can optimize multiple properties at once.

I am leaning towards an accept for this paper, since not only does the technique presented seem general, the authors does in depth analysis into the model and how it affects drug discovery.
- Recursive black box translation seems to be widely applicable to new models.
- The model seems to reach a significantly better state of the art on the metrics proposed.
- None of the baselines seem to use SELFIES as the string of choice.
  This means it's difficult to tell how much the "Blackbox recursive" part of the algorithm adds to the model.
  An ablation experiment without BBRT might inform us of how much of the benefit is due to the molecule representation (Fig 4A reports the mean, but it would be good to have the same metric as Table 1).
- The authors provide an in depth discussion about how having molecular traces would hhelp in drug design.
  This makes the tool seem more widely appealing and useful.

A few questions would clear up the strengths of the paper:
- Is there a connection to the backtranslation work in Lample 2018? (Phrase-Based & Neural Unsupervised Machine Translation)
  It seems like a similar idea - except in this domain, the target language and source language are the same.
- How can there be multiple scoring functions?
  Were they combined in one run, or were these separately optimized runs? Are these only used in Figure 4?
- Why would beam search do less well than stochastic?
  Is it because during recursive translation, the beam search variants have low diversity?
  Then, training with stochastic decoding and generation with a beam search should do even better, right?
  This would highlight that the advantage of stochastic decoding is really online in the context of recursive translation, not generally.
- What is the point of Fig 4A right? Why do we expect that maximizing non-logP properties will increase mean logP?


**Experience Assessment:**

I have read many papers in this area.

**Review Assessment: Checking Correctness Of Derivations And Theory:**

N/A

**Review Assessment: Checking Correctness Of Experiments:**

I assessed the sensibility of the experiments.

**Review Assessment: Thoroughness In Paper Reading:**

I read the paper at least twice and used my best judgement in assessing the paper.

---

> ### Author Response · Authors · 2019-11-15
> **Response Part 1**
>
> We thank the reviewer for their thoughtful response.
>
> > "I am leaning towards an accept for this paper, since not only does the technique presented seem general, the authors does in depth analysis into the model and how it affects drug discovery.
> - Recursive black box translation seems to be widely applicable to new models.
> - The model seems to reach a significantly better state of the art on the metrics proposed.
> - None of the baselines seem to use SELFIES as the string of choice."
>
> The Seq2Seq baseline was trained on SELFIES while JT-VAE, GCPN, and VJTNN are graph-based. We use the reported numbers from ORGAN, which was trained on SMILES strings. We agree a better comparison to ORGAN as a baseline would be trained on SELFIES.
>
> > "This means it's difficult to tell how much the "Blackbox recursive" part of the algorithm adds to the model. An ablation experiment without BBRT might inform us of how much of the benefit is due to the molecule representation (Fig 4A reports the mean, but it would be good to have the same metric as Table 1)."
>
> This ablation experiment is reported. In Table 1, the Seq2Seq baseline is trained on SELFIES without BBRT. Figure 2 compares the top 100 generated compounds according to Seq2Seq vs BBRT-Seq2Seq and JTNN vs BBRT-JTNN controlling for the molecular representation.
>
> > "A few questions would clear up the strengths of the paper:
> - Is there a connection to the backtranslation work in Lample 2018? (Phrase-Based & Neural Unsupervised Machine Translation)
>   It seems like a similar idea - except in this domain, the target language and source language are the same."
>
> We acknowledge that there are some philosophical similarities between our work and the iterative backtranslation (Sennrich et al. 2015) in Lample et al. 2018. Some key distinctions:
>
> 1. The design goal and motivation are different: while backtranslation (Sennrich et al. 2015) often is used to improve learning in low-resource settings by construction of augmented training sets, ours focuses on the dynamics at test-time; by construction and in this paper, our training data already consists of molecular pairs $(X, Y)$  with high structural similarity $\tau$, for a given property. We do note that an interesting direction for exploring constrained optimization would be where we want to learn a prior over training pairs with a relaxed constraint and use this for a more stringent, high-constraint setting (a “low resource” setting, where we might encounter a paucity of data due to expensive experiments).
>
> 2. The Lample et al. 2018 uses two different language models are learned over source $s$ and target $t$ languages  $(P_{s \rightarrow t}$ and $P_{t \rightarrow s})$. The source-to-target model is applied to source sequences to generate inputs for training the target-to-source model. The backtranslation step generates source and target sequences using the learned translation models $P^{(k-1)}_{t \rightarrow s}$ and $P^{(k-1)}_{s \rightarrow t}$ and then trains new translation models $P^{(k)}_{t \rightarrow s}$ and $P^{(k)}_{s \rightarrow t}$ using the sequences. We use only one model, and after $n$ recursive iterations, we ensemble the generated sequences whilst scoring on a desired objective. We acknowledge that, in principle, BBRT could be extended to ensembling recursive outputs from more than one model.
>
>
> > "- How can there be multiple scoring functions?
>   Were they combined in one run, or were these separately optimized runs?"
>
> Separate runs.
>
> > "Are these only used in Figure 4?"
>
> Fig. 4 shows results from 4 scoring functions and Fig. 6 compares 2. Fig. 5 uses the max pairwise sim scoring function while the remaining results use the same scoring function that was used to optimize compounds. Finally, the top reported compounds reported in Table 1 and Fig. 3 are from aggregating results across decoding strategies and scoring functions.
>
> > "- Why would beam search do less well than stochastic?
>   Is it because during recursive translation, the beam search variants have low diversity?"
>
> Finding sequences that have high probability under the model (when trained with teacher forcing) is not always well-calibrated for downstream tasks (like finding sequences that score highly on chemical properties). This is a well-known phenomena in the NLP literature. Because stochastic decoding can generate a larger number of diverse samples, empirically we have found that after generating a large number of samples (say 100) we are able to find higher scoring compounds relative to the sequence that has the highest log probability under the model (found using beam search).
>
> > "Then, training with stochastic decoding and generation with a beam search should do even better, right?"
> Yes, training with stochastic decoding and beam search does do better than just stochastic decoding.

---

> ### Author Response · Authors · 2019-11-15
> **Response Part 2**
>
> > "This would highlight that the advantage of stochastic decoding is really online in the context of recursive translation, not generally."
>
> Stochastic decoding certainly helps for recursive translations. However, we also observed that it helps generally for non-recursive translations. In Fig. 4A left, the dotted lines show mean logP for three decoding strategies (beam search, top 2 and top 5 sampling) using a Seq2Seq model. We observe that beam search generates the lowest mean logP.
>
>
> > "- What is the point of Fig 4A right? Why do we expect that maximizing non-logP properties will increase mean logP?"
>
> This is a plot comparing four scoring functions. The translation model is trained to optimize logP for all lines. The difference is how the intermediate outputs are ranked and thus which top k molecules are propagated forward. We expect scoring by non-logP will increase mean logP because the translation still optimizes for logP. Among the optimized logP candidates, the scoring function ranks these choices by one of four scoring functions.

---

### Decision · Program_Chairs · 2019-12-19

**Decision:**

Reject

**Comment:**

This paper presents a simple method for improving molecular optimization with a learned model. The method operates by repeatedly feeding generated molecules back through an encoder decoder pair trained to maximize a desired property. Reviewers liked the simplicity of the method, and found it interesting but ultimately there were concerns about the metrics used to evaluate the method. Reviewers 3 and 4 both noted issues with the log P (and penalized log P) metric, noting that it is possible to artificially increase both metrics in a way that isn't useful in practice. During the discussion phase, Reviewer 4 constructed a specific example where simply adding long carbon chains to a molecule would yield a linear increase the penalized log P metric, and noted that the "best molecules" found by the method in Figure 3 also have extremely long carbon chains (long carbon chains are not generally desirable for drug discovery).
I recommend the authors resubmit after finding a better way to evaluate that their method generates molecules with more useful properties for drug discovery.